# BulletTrain: Accelerating Robust Neural Network Training via Boundary Example Mining

**Weizhe Hua[1], Yichi Zhang[1], Chuan Guo[2], Zhiru Zhang[1], G. Edward Suh[1,2]**
[1]Cornell University, [2]Facebook AI Research
{wh399,yz2499,zhiruz,gs272}@cornell.edu, chuanguo@fb.com

## Abstract

Neural network robustness has become a central topic in machine learning in recent years. Most training algorithms that improve the model's robustness to adversarial and common corruptions also introduce a large computational overhead, requiring as many as *ten times* the number of forward and backward passes in order to converge. To combat this inefficiency, we propose BulletTrain — a boundary example mining technique to drastically reduce the computational cost of robust training. Our key observation is that only a small fraction of examples are beneficial for improving robustness. BulletTrain dynamically predicts these important examples and optimizes robust training algorithms to focus on the important examples. We apply our technique to several existing robust training algorithms and achieve a $2.2\times$ speed-up for TRADES and MART on CIFAR-10 and a $1.7\times$ speed-up for AugMix on CIFAR-10-C and CIFAR-100-C *without any reduction* in clean and robust accuracy.

## 1 Introduction

In the past decade, the performance and capabilities of deep neural networks (DNNs) have improved at an unprecedented pace. However, the reliability of DNNs is undermined by their lack of robustness against common and adversarial corruptions, raising concerns about their use in safety-critical applications such as autonomous driving (Amodei et al., 2016). Towards improving the robustness of DNNs, a line of recent works proposed robust training by generating and learning from corrupted examples in addition to clean examples (Cubuk et al., 2019; Ding et al., 2020; Hendrycks et al., 2020; Madry et al., 2018; Wang et al., 2020; Zhang et al., 2019b). While these methods have demonstrably enhanced model robustness, they are also very costly to deploy since the overall training time can be increased by up to ten times.

Hard negative example mining techniques have shown success in improving the convergence rates of support vector machine (SVM) (Joachims, 1999), DNNs (Shrivastava et al., 2016), metric learning (Manmatha et al., 2017), and contrastive learning (Kalantidis et al., 2020). Particularly, SVM with hard example mining maintains a working set of "hard" examples and alternates between training an SVM to convergence on the working set and updating the working set, where the "hard" examples are significant in that they violate the margins of the current model. Partly inspired by hard example mining, we hypothesize that there exists a small subset of examples that is critical to improving the robustness of the DNN model, so that reducing the computation of the remaining examples leads to a significant reduction in run time without compromising the robustness of the model.

As depicted in Figure 1b, the essential idea is to predict and focus on the "important" subset of clean samples based on their geometric distance to the decision boundary (i.e., margin) of the current model. For example, the correctly classified clean examples with a large margin are unlikely to help improving robustness as the corrupted examples generated from them are still likely to be appropriately categorized by the model. We refer to "important" examples as boundary examples since they are close to the decision boundary (e.g., □ in Figure 1b). After identifying the boundary

| 1st iteration | 2nd iteration | 3rd iteration |
|---|---|---|

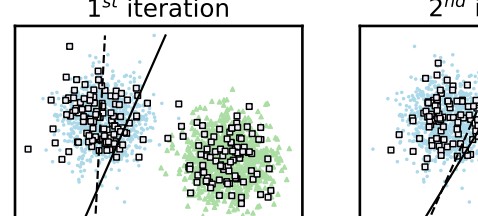 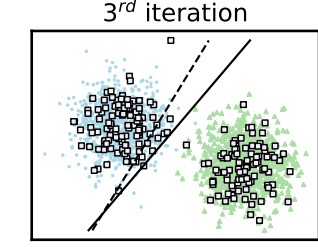

(a) Existing robust neural network training algorithms — Standard robust training approaches generate corrupted samples with equal effort for all samples (□) in a mini-batch.

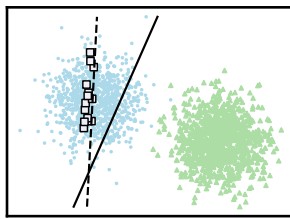 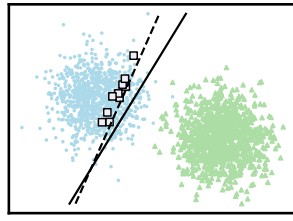 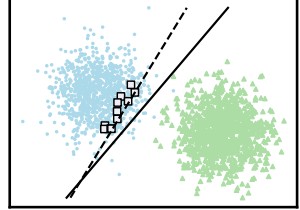

(b) BulletTrain — The proposed scheme focuses on a small subset of "important" clean samples (□) and allocates more computational cost on obtaining corrupted samples for those important ones. This figure illustrates the case that only generates the corrupted samples for the important samples.

Figure 1: Comparison between existing robust training approaches and BulletTrain on a synthetic binary classification task — The dash line is the current decision boundary of the linear model and the solid line represents the updated decision boundary after taking a SGD step.

examples, the computational effort on generating corrupted samples can be reduced for the rest of the clean examples, thus accelerating the overall robust training process.

In this work, we propose BulletTrain[1], a dynamic optimization that exploits boundary example mining to reduce the computational effort of robust DNN training. BulletTrain applies to robust training against either adversarial or common corruptions. It can substantially speed up state-of-the-art algorithms (Hendrycks et al., 2020; Wang et al., 2020; Zhang et al., 2019b) with almost no loss in clean and/or robust accuracy. Figure 1 visualizes the situation where BulletTrain generates corrupted examples only for a small fraction of clean examples, while a standard robust training scheme requires obtaining corrupted examples for all examples.

To the best of our knowledge, BulletTrain is the first work to exploit the idea of hard example mining in the context of accelerating robust DNN training. Unlike existing approaches which are limited to reducing the compute for all examples uniformly, BulletTrain distributes the compute for clean examples differently, focusing on a small set of boundary examples. BulletTrain is also generally applicable to robust training against common and adversarial corruptions. The end result is that BulletTrain can lower the running time of the adversarial training such as TRADES (Zhang et al., 2019b) and MART (Wang et al., 2020) by more than $2.1\times$ on CIFAR-10 (Krizhevsky, 2009) and the robust training against common corruptions such as AugMix (Hendrycks et al., 2020) by $1.7\times$ on CIFAR-10-C (Hendrycks and Dietterich, 2019) without compromising the clean and robust accuracy.

## 2 Robust Training Overview

**Preliminaries.** We focus on K-class ($K \geq 2$) classification problems. Given a dataset $\{(\mathbf{x}_i, y_i)\}_{i=1,...,n}$ with a clean example $\mathbf{x}_i \in \mathbb{R}^d$ and its associated ground truth label $y_i \in \{1, ..., K\}$, a DNN classifier $h_\theta$ parameterized by $\theta$ generates the logits output $h_\theta(\mathbf{x}_i)$ for each class, where $h_\theta(\mathbf{x}_i) = (h_\theta^1(\mathbf{x}_i), ..., h_\theta^K(\mathbf{x}_i))$. The predicted label $\hat{y}_i$ can be expressed as $\hat{y}_i = \arg\max_k h_\theta^k(\mathbf{x}_i)$.

To improve the robustness of a DNN classifier, state-of-the-art approaches (Hendrycks and Dietterich, 2019; Madry et al., 2018; Wang et al., 2020; Zhang et al., 2019b) generate corrupted inputs $\mathbf{x}_i'$ for the corresponding clean example $\mathbf{x}_i$ at training time and minimize the loss of the classifier with respect to the corrupted input. For robustness against adversarial attacks, $\mathbf{x}_i'$ can be obtained using

---

[1]The name of BulletTrain implies robust (bullet-proof) and high-speed training.

**Algorithm 1:** Standard robust DNN training algorithm.

---
**Input** : batch size $m$, a $N$-step generation function $G_N$ for generating corrupted samples
1 **while** $\theta$ not converged **do**
2      Read mini-batch $\{\mathbf{x}_1, ..., \mathbf{x}_m; y_1, ..., y_m\}$ from training set
3      **for** $i = 1, ..., m$ **do**
4         $\mathbf{x}'_i \leftarrow G_N(\mathbf{x}_i)$                 // $G_N(\cdot)$ is an $N$-step procedure for generating perturbations
5      **end**
6      $\theta \leftarrow \theta - \nabla_\theta(\mathcal{L}(\{h_\theta(\mathbf{x}_1), ..., h_\theta(\mathbf{x}_m)\}, \{h_\theta(\mathbf{x}'_1), ..., h_\theta(\mathbf{x}'_m)\}))$      // $\mathcal{L}$ is the surrogate loss
7 **end**

---

adversarial attack methods such as projected gradient descent (PGD) (Madry et al., 2018). For robustness to common corruptions and perturbations, $\mathbf{x}'_i$ is generated by introducing different data augmentations (Hendrycks and Dietterich, 2019). Algorithm 1 shows pseudo-code for a standard robust training algorithm.

**Computational cost of robust training.** Robust DNN training algorithms such as Algorithm 1 improve model robustness at the cost of paying extra computational overhead. There are two main sources of the overhead: 1) The procedure $G_N$ for generating corrupted input; 2) Forward and backward propagate using the surrogate loss $\mathcal{L}$ with respect to both clean and corrupted samples. We denote the extra computational cost for robust neural network training as $C[G_N, \mathcal{L}]$. For adversarial robustness, PGD adversarial training generates the adversarially corrupted input by solving the inner maximization $\max_{\delta \in \Delta} \mathcal{L}(\mathbf{x} + \delta, y; \theta)$ approximately using an $N$-step projected gradient descent method , where $\Delta$ is the perceptibility threshold and is usually defined using an $l_p$ distance metric: ($\Delta = \{\delta : \|\delta\|_p < \epsilon\}$ for some $\epsilon \geq 0$). As a result, PGD adversarial training is roughly $N$ times more expensive than ordinary DNN training, where $N$ is usually between 7 and 20.

In addition, Hendrycks et al. (2020) propose to boost the robustness against common corruptions using Jensen-Shannon divergence (JSD) as the surrogate loss. JSD loss can be computed by first generating two corrupted examples ($\mathbf{x}'$ and $\mathbf{x}''$), calculating a linear combination of the logits as $M = (h_\theta(\mathbf{x}) + h_\theta(\mathbf{x}') + h_\theta(\mathbf{x}''))/3$, and then computing $\mathcal{L}(\mathbf{x}, \mathbf{x}', \mathbf{x}'') = \frac{1}{3}(\mathrm{KL}[h_\theta(\mathbf{x}) \parallel M] + \mathrm{KL}[h_\theta(\mathbf{x}') \parallel M] + \mathrm{KL}[h_\theta(\mathbf{x}'') \parallel M])$ , where KL stands for the Kullback–Leibler divergence. As a result, the surrogate loss for improving the robustness against common corruptions triples the cost of the ordinary training as it requires forward and backward propagation for both $\mathbf{x}, \mathbf{x}'$, and $\mathbf{x}''$.

**Existing efficient training algorithms.** A number of recent efforts (Andriushchenko and Flammarion, 2020; Shafahi et al., 2019; Wong et al., 2020) aimed at reducing the complexity of adversarial example generation, which constitutes the main source of computational overhead in adversarial training. For instance, Wong et al. (2020) proposed a variant of the single-step fast gradient sign method (FGSM) in place of multi-step methods, which drastically reduced the computational cost. However, it was later shown that the single-step approaches fail catastrophically against stronger adversaries (Andriushchenko and Flammarion, 2020). FGSM-based methods can also cause substantial degradation in robust accuracy (Chen et al., 2020) for recently proposed adversarial training methods such as TRADES (Zhang et al., 2019b). This result suggests that the adversarial examples generated using single-step FGSM are not challenging enough for learning a robust classifier.

In addition, existing efficient training approaches cannot be generalized to robust training against common corruptions. The state-of-the-art robust training algorithms against common corruptions (Cubuk et al., 2019; Hendrycks et al., 2020) use data augmentations to create corrupted samples in a single pass, where the number of iterations can not be further reduced.

## 3 BulletTrain

### 3.1 Motivation

As discussed in Section 2, robust training can lead to a significant slowdown compared to ordinary DNN training. To reduce the overhead of robust training, our key insight is that different inputs contribute unequally to improving the model's robustness (which we later show), hence we can dynamically allocate different amounts of computation (i.e., different $N$) for generating corrupted samples depending on their "importance". In detail, we first categorize the set of all training samples into a disjoint union of three sets — outlier, robust, and boundary examples — as follows:

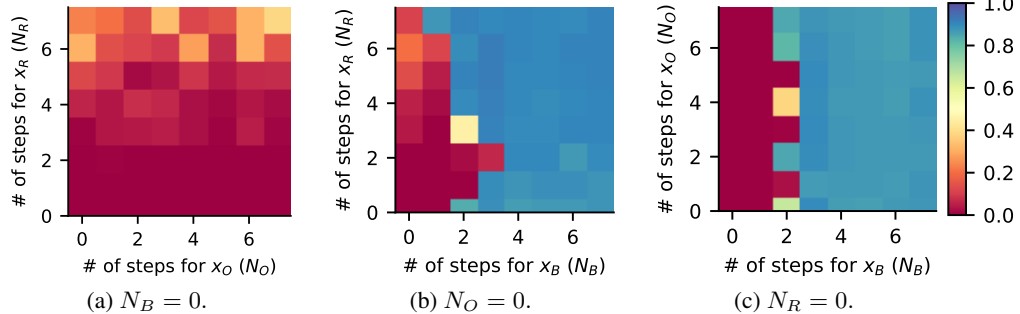

(a) $N_B = 0$.     (b) $N_O = 0$.     (c) $N_R = 0$.

Figure 2: The robust accuracy of setting one of $N_B$, $N_R$, and $N_O$ to be zero and varying the other two between zero and seven on MNIST — All experiments are performed on a four layer convolutional neural network trained with ten epochs. The step size for each example is set to $1.7\epsilon/N$. The robust accuracy represented by color is evaluated using PGD attacks with 100 steps and 20 restarts.

- Outlier examples $\mathbf{x}_O$: the clean input $\mathbf{x}$ is misclassified by the DNN model $h_\theta$.

$$\mathbf{x}_O = \{\mathbf{x} \mid \arg\max_{k=1,\dots,K} h_\theta^k(\mathbf{x}) \neq y\} \tag{1}$$

- Boundary examples $\mathbf{x}_B$: the clean input $\mathbf{x}$ is correctly classified while the corresponding corrupted input $\mathbf{x}'$ is misclassified by the DNN model $h_\theta$.

$$\mathbf{x}_B = \{\mathbf{x} \mid \arg\max_{k=1,\dots K} h_\theta^k(\mathbf{x}) = y \wedge \arg\max_{k=1,\dots,K} h_\theta^k(\mathbf{x}') \neq y\} \tag{2}$$

- Robust examples $\mathbf{x}_R$: both the clean input $\mathbf{x}$ and the corresponding corrupted input $\mathbf{x}' = G_N(\mathbf{x})$ are correctly classified by the DNN model $h_\theta$.

$$\mathbf{x}_R = \{\mathbf{x} \mid \arg\max_{k=1,\dots K} h_\theta^k(\mathbf{x}) = \arg\max_{k=1,\dots,K} h_\theta^k(\mathbf{x}') = y\} \tag{3}$$

Note that our definition of the outlier, boundary, and robust examples is dependent on the current model $h_\theta$ and can dynamically shift during training. That is, an outlier example may become a boundary example later on when the robustness of the model improves. Similarly, a robust example can also become a boundary or even outlier example when robustness improves in other regions of the training distribution.

Robust training algorithms improve the robustness of a DNN model by increasing the classification margin for each training sample $\mathbf{x}$ so that its distance to the corresponding corrupted sample (i.e., $\|\mathbf{x} - \mathbf{x}'\|_p$) is smaller than the margin. We hypothesize that this per-sample margin is an indicator of the amount of contribution that provided by this sample during robust training.

- **Outlier examples** $\mathbf{x}_O$ have a negative margin because they are misclassified by the current DNN model. Therefore, it is unnecessary to generate even "harder" corrupted examples and optimize the model using these "harder" examples.
- **Boundary examples** $\mathbf{x}_B$ have a positive margin to the current decision boundary but can be perturbed to cause misclassification. Training on a corrupted version of these samples is most helpful to increase the margin, thus making the model more robust.
- **Robust examples** $\mathbf{x}_R$ already have a margin large enough to tolerate common or adversarial corruption. As a result, training on corrupted versions of these examples would provide less benefit compared to training on corrupted boundary examples.

To test our hypothesis, we conduct an empirical study of PGD adversarial training against $l_\infty$ attack ($\epsilon = 0.3$) on MNIST (LeCun et al., 2010) in a leave-one-out manner. The baseline PGD training achieves 90.6% robust accuracy by setting the number of steps for all samples to be seven. In our experiment, we fix the number of PGD iterations ($N$) for one of $\mathbf{x}_B$, $\mathbf{x}_O$, and $\mathbf{x}_R$ to be zero while varying $N$ between zero and seven for the other two classes of data. The separation between $\mathbf{x}_B$, $\mathbf{x}_O$, and $\mathbf{x}_R$ for each mini-batch of data is oracle, which is determined using ten PGD steps. As depicted in Figure 2a, PGD training fails to achieve a robust accuracy over 40% when the number of steps for $\mathbf{x}_B$ ($N_B$) is zero; thus demonstrating the importance of $\mathbf{x}_B$. When $N_O$ equals zero (Fig. 2b), the highest robust accuracy is 1% higher than that of the baseline, indicating that setting $N_O$ to zero does

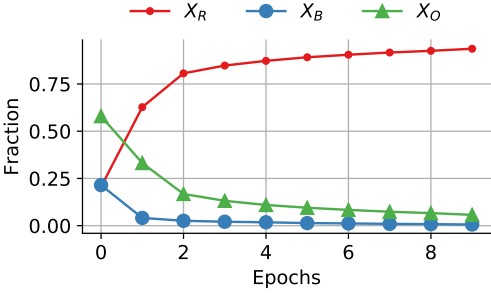

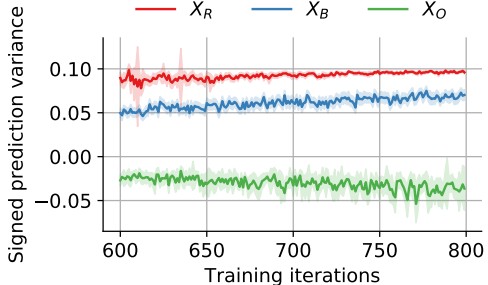

Figure 3: The fraction of $\mathbf{x}_R$, $\mathbf{x}_B$, and $\mathbf{x}_O$ over epochs on MNIST.

Figure 4: The average signed prediction variance of $\mathbf{x}_R$, $\mathbf{x}_B$, and $\mathbf{x}_O$ over different iterations.

not impair the effectiveness of PGD training. Lastly, Figure 2c shows that PGD training can achieve a reasonable robust accuracy without paying the extra computational cost on $\mathbf{x}_R$. However, the highest robust accuracy achieved with this setup is $89.2\%$, which is $1.4\%$ lower than the baseline. Therefore, setting a non-zero $N_R$ might help improving the robustness of the model in practice.

This empirical observation reveals an opportunity to accelerate robust DNN training by focusing on the boundary examples. In doing so, the overhead of robust DNN training is then determined by the proportion of boundary examples among all training samples. As shown in Figure 3, the fraction of $\mathbf{x}_R$ increases while the fraction of $\mathbf{x}_O$ decreases over epochs as the model becomes more robust. Moreover, the fraction of $\mathbf{x}_B$ is relatively small compared to the other two classes of samples since boundary examples only present within the $\epsilon$ vicinity of the decision boundary. Therefore, we believe that the computational cost of robust training ($C[G_N, \mathcal{L}]$) can be reduced substantially by attending more to the boundary examples.

### 3.2 Algorithm

**Separate the clean samples.** First note that outlier examples can be easily separated from the other two categories because the current model misclassifies them. Thus, we use the sign of prediction to identify outliers:

$$\text{Sign of prediction} = \begin{cases} +1, & \text{if } \arg\max_{k=1,\ldots K} h_\theta^k(\mathbf{x}) = y \\ -1, & \text{otherwise} \end{cases} \tag{4}$$

We can further distinguish between the boundary and robust examples by exploiting the fact that boundary examples are closer to the decision boundary than the robust ones. As suggested by prior work on importance sampling (Chang et al., 2017), the prediction variance ($\text{Var}[\text{Softmax}(h_\theta(\mathbf{x}))]$) can be used to measure the uncertainty of samples in a classification problem, where $\text{Var}[\cdot]$ and $\text{Softmax}(\cdot)$ stand for variance and softmax function respectively. Samples with low uncertainty are usually farther from the decision boundary than samples with high uncertainty. In other words, the prediction variance indicates the distance between the sample and the decision boundary, which can be leveraged to separate $\mathbf{x}_B$ and $\mathbf{x}_R$. We multiply the sign of prediction and the prediction variance as the final metric — **signed prediction variance** ($\text{SVar}[\cdot]$). As shown in Figure 4, the signed prediction variances of $\mathbf{x}_R$, $\mathbf{x}_B$, and $\mathbf{x}_O$ from different mini-batches are mostly rank correlated ($\text{SVar}[\mathbf{x}_O] \leq \text{SVar}[\mathbf{x}_B] \leq \text{SVar}[\mathbf{x}_R]$), suggesting that the proposed metric can distinguish between different types of samples. We empirically verified that other uncertainty measures such as cross-entropy loss and gradient norm (Katharopoulos and Fleuret, 2018) perform comparably to prediction variance, and hence use the signed prediction variance in this study.

As BulletTrain aims to reduce the computational cost, it requires a lightweight approach to differentiate between samples. The outliers can be separated easily based on the sign of SVar. To separate between $\mathbf{x}_B$ and $\mathbf{x}_R$ with minimal cost, we propose to distinguish the two categories of samples by estimating the fraction of $\mathbf{x}_B$ or $\mathbf{x}_R$. The fractions of $\mathbf{x}_R$ and $\mathbf{x}_B$ cannot be computed directly without producing the corrupted samples. Therefore, we estimate the fraction of $\mathbf{x}_R$ ($F_R$) of the current mini-batch using an exponential moving average of robust accuracy from previous mini-batches. As the robust accuracy of the model should be reasonably consistent across adjacent mini-batches, the moving average of robust accuracy from the past is a good approximation of $F_R$ of the current

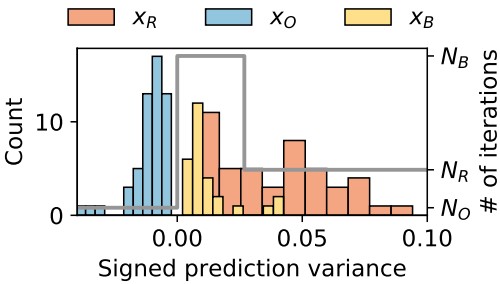
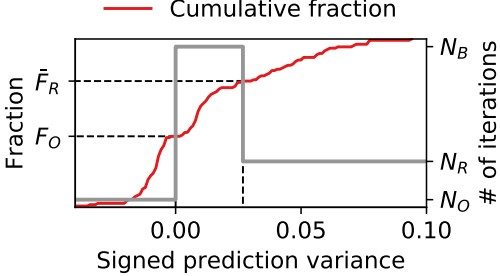

(a) The distribution of signed prediction variance of clean samples in a mini-batch.

(b) The number of steps for each sample is assigned based on the estimated fractions $F_O$ and $\bar{F}_R$.

Figure 5: Illustration of the compute allocation scheme — The grey line shows the number of steps assigned for the clean samples in a mini-batch.

---

**Algorithm 2:** BulletTrain.

---

**Input** : batch size $m$, a process for generating corrupted samples $G_N$, classification rule $f$, momentum $p$

8    $F_O \leftarrow 0; F_R \leftarrow 0$

9    **while** $\theta$ not converged **do**

10      Read mini-batch $\{\mathbf{x}_1, ..., \mathbf{x}_m; y_1, ..., y_m\}$ from training set

11      $\{\mathbf{z}_1, ..., \mathbf{z}_m\} \leftarrow \{h_\theta(\mathbf{x}_1), ..., h_\theta(\mathbf{x}_m)\}$

12      $\mathbf{x}_R, \mathbf{x}_B, \mathbf{x}_O \leftarrow f(\{\mathbf{x}_1, ..., \mathbf{x}_m\}, F_R)$            // Separate between examples

13      $\mathbf{x}'_O \leftarrow \boldsymbol{G_{N_O}}(\mathbf{x}_O), \mathbf{x}'_R \leftarrow \boldsymbol{G_{N_R}}(\mathbf{x}'_R), \mathbf{x}'_B \leftarrow \boldsymbol{G_{N_B}}(\mathbf{x}'_B)$        // Produce corrupted examples

14      $\{\mathbf{x}'_1, ..., \mathbf{x}'_m\} \leftarrow \mathbf{x}'_B \parallel \mathbf{x}'_R \parallel \mathbf{x}'_O; \{\mathbf{z}'_1, ..., \mathbf{z}'_m\} \leftarrow \{h_\theta(\mathbf{x}'_1), ..., h_\theta(\mathbf{x}'_m)\}$

15      $F_R \leftarrow p \cdot F_R + (1 - p) \cdot \gamma \cdot \sum_i^m \mathbb{1}(\arg\max_k(\mathbf{z}'_i)^k = y_i)/m$        // Update $F_R$

16      $\theta \leftarrow \theta - \nabla_\theta(\mathcal{L}(\{\mathbf{z}_1, ..., \mathbf{z}_m\}, \{\mathbf{z}'_1, ..., \mathbf{z}'_m\}))$

17 **end**

---

mini-batch. In addition, Figure 5a shows that $\mathbf{x}_B$ and $\mathbf{x}_R$ are not completely separable using the signed prediction variance. To make conservative predictions for robust examples, we scale the $F_R$ with a scaling factor $\gamma$, where $\gamma \in (0, 1]$. $F_R$ can be obtained by:

$$F_R \leftarrow p \cdot F_R + (1 - p) \cdot \gamma \cdot \frac{\sum_i^m \mathbb{1}(\arg\max_k h_\theta^k(\mathbf{x}'_i) = y_i)}{m} \tag{5}$$

where $\mathbb{1}(\cdot)$ is the indicator function, $\mathbf{x}'$ are the corrupted samples in the previous mini-batch, $p$ is the momentum, and $F_R$ is initialized to be zero.

**Compute allocation scheme.** After dividing the clean examples into three sets, it is relatively straightforward to assign computational effort to the samples in each category. Following our intuition discussed in Section 3.1, we spend the largest number of steps ($N_B$) on the samples predicted as boundary examples. For the rest of the samples, we assign the second largest number of steps ($N_R$) to robust examples and the lowest number of steps ($N_O$) to outliers.

Given $F_R$, we first compute the signed prediction variances of all clean examples as $\text{SVar}_1^m = \{\text{SVar}[h_\theta(\mathbf{x}_1)], ..., \text{SVar}[h_\theta(\mathbf{x}_m)]\}$ and then formulate non-parametric classification rule ($f$) as:

$$f(\mathbf{x}_i, F_R) = \mathbf{x}_i \in \begin{cases} \mathbf{x}_O, & \text{if } \text{SVar}[h_\theta(\mathbf{x}_i)] < 0 \\ \mathbf{x}_B, & \text{if } 0 \le \text{SVar}[h_\theta(\mathbf{x}_i)] < \text{Percentile}(\text{SVar}_1^m, \bar{F}_R) \\ \mathbf{x}_R, & \text{otherwise} \end{cases} \tag{6}$$

where $\bar{F}_R$ equals to $1 - F_R$ and $\text{Percentile}(\mathbf{X}, q)$ returns the $q^{th}$ percentile of $\mathbf{X}$. The computational cost of $f$ comes mostly from obtaining the logits of clean example $h_\theta(\mathbf{x})$, which is already computed in robust DNN training. Therefore, the additional cost for assigning compute for each sample is negligible. Figure 5 (grey line) visualizes the compute allocation scheme for a mini-batch of samples. It is worth noting that the proposed $f$ resembles a discretized Gaussian function, suggesting that a more refined allocation scheme can be obtained from a quantized Gaussian function. Here, we present the full algorithm of BulletTrain in Algorithm 2.

BulletTrain first identifies the robust, boundary, and outlier samples during training time by applying the classification rule $f$ to signed prediction variance. The separation is relatively accurate as the

Table 1: Robust accuracy and speedup of applying BulletTrain (BT) to robust neural network training against common corruptions (AugMix) on CIFAR-10 and CIFAR-100.

| Defenses | CIFAR-10 | | | CIFAR-100 | | |
|---|---|---|---|---|---|---|
| | Robust Acc. | $\bar{F}_B$ | Theor. Speedup | Robust Acc. | $\bar{F}_B$ | Theor. Speedup |
| AugMix | 88.8% | 1 | 1× | 64.0% | 1 | 1× |
| AugMix − JSD loss | 86.9% | - | 3× | 60.2% | - | 3× |
| AugMix + BT | **88.8%** | 0.35 | 1.8× | **63.6%** | 0.36 | 1.8× |

signed prediction variance can capture the margin between samples and the decision boundary of the current model. Then, clean examples take either $N_O$, $N_R$, or $N_B$ steps to generate the corresponding corrupted samples. Since only a small fraction of samples (i.e., boundary examples) requires executing the same number of iterations ($N_B$) as the baseline, the computational cost is reduced significantly. Lastly, the surrogate loss is computed with respect to the clean and corrupted samples. If both $N_R$ and $N_O$ are equal to zero, the corrupted sample remains the same as the original sample, thus eliminating the additional cost of forward and backward propagation of the corrupted samples. For simplicity, *we let $N_B$ be the same as $N$ in the original robust training, $N_O$ be zero, and $N_R \in [N_O, N_B)$.*

In adversarial training, since PGD-based algorithms typically employ a large number of steps, the main saving from applying BulletTrain stems from reducing the number of steps for outlier and robust examples. In training against common corruptions, the reduction in computation comes mainly from not computing surrogate loss and gradient for corrupted samples. For example, the computational overhead of AugMix can be reduced by half if we only generate corrupted samples for half of the clean samples. We will show that this leads to a large reduction in training overhead in Section 4.

## 4 Experiments

**Experimental setup.** To demonstrate the efficacy and general applicability of the proposed scheme, we leverage BulletTrain to accelerate TRADES (Zhang et al., 2019b) and MART (Wang et al., 2020) against adversarial attacks on CIFAR-10 and AugMix (Hendrycks et al., 2020) against common corruptions on CIFAR-10-C and CIFAR-100-C. To make a fair comparison, we adopt the same threat model employed in the original paper and measure the robustness of the same network architectures. Specifically, we evaluate the robustness against common corruptions of WideResNet-40-2 (Zagoruyko and Komodakis, 2016) using corrupted test samples in CIFAR-10-C and CIFAR-100-C. When we apply BulletTrain to AugMix, we set $N_O = N_R = 0$ and $N_B = 1$ as it only allows to turn on or off the AugMiX data augmentation for clean samples. The adversarial robustness of WideResNet-34-10 is evaluated using PGD[20] (i.e., PGD attack with 20 iterations) $l_\infty$ attack for TRADES and MART. The perturbation ($\epsilon$) of the $l_\infty$ attacks on CIFAR-10 is set to be 0.031. Both TRADES and MART without BulletTrain are trained using 10-step PGD with a step size $\alpha = 0.007$. When BulletTrain is applied, we set $N_B = 10$ with $\alpha = 0.007$, $N_R \in [0, 2]$ with $\alpha = 1.7\epsilon/N_R$, $N_O = 0$, and $\gamma = 0.8$.

We use the measured average fraction of $\mathbf{x}_B$, $\mathbf{x}_R$, and $\mathbf{x}_O$ (i.e., $\bar{F}_B$, $\bar{F}_R$, and $\bar{F}_O$) to calculate the theoretical speedup of BulletTrain, assuming that all computations without dependencies can be parallelized with unlimited hardware resources. Specifically, in adversarial training, the $N$ iterations for generating destructive examples and one iteration for updating the model must be executed sequentially. Therefore, the theoretical speedup can be written as follows:

$$\text{Theoretical speedup} = \frac{N + 1}{\bar{F}_B \cdot N_B + \bar{F}_R \cdot N_R + \bar{F}_O \cdot N_O + 1} \qquad (7)$$

**Reduction in computation cost.** As listed in Table 1, BulletTrain can reduce the computational cost of AugMix by 1.78× and 1.75× with no and 0.4% robust accuracy drop on CIFAR-10 and CIFAR-100, respectively. The compute savings stem from only selecting 35% of clean examples to perform the JSD loss. Compared to removing the JSD loss completely for all examples, BulletTrain improves the robust accuracy by 1.9% and 3.4% on CIFAR-10 and CIFAR-100, respectively.

Table 2 compares clean and robust accuracy of TRADES with and without BulletTrain. Compared to the original TRADES adversarial training (TRADES$_{\lambda=1/6}$), BulletTrain with $N_R = 2$ achieves a 0.3% lower robust accuracy and a 0.9% higher clean accuracy. It is worth noting that the robustness

Table 2: Robust accuracy and speedup of applying BulletTrain (BT) to TRADES adversarial training against the white-box $\text{PGD}^{20}$ attack on CIFAR-10 — $\epsilon = 0.8$ is used for TRADES with BT.

| Defenses | Clean Acc. | Robust Acc. | $\bar{F}_B$ | $\bar{F}_R$ | $\bar{F}_O$ | Theor./Wall-clock Speedup |
|---|---|---|---|---|---|---|
| $\text{TRADES}_{\lambda=1}$ | 88.64% | 49.14% | 1 | 0 | 0 | $1\times$ / $1\times$ |
| $\text{TRADES}_{\lambda=\frac{1}{6}}$ | 84.92% | 56.61% | 1 | 0 | 0 | $1\times$ / $1\times$ |
| $\text{TRADES}_{\lambda=\frac{1}{6}}$ + FAST | 93.94% | 4.48% | 1 | 0 | 0 | $5.5\times$ / $3.7\times$ |
| $\text{TRADES}_{\lambda=1}$ + YOPO-2-5 | 88.47% | 45.28% | 1 | 0 | 0 | - / $3.1\times$ |
| $\text{TRADES}_{\lambda=\frac{1}{6}}$ + YOPO-2-5 | 89.95% | 46.60% | 1 | 0 | 0 | - / $3.1\times$ |
| $\text{TRADES}_{\lambda=\frac{1}{6}}$ + $\text{BT}_{N_R=0}$ | 87.50% | 52.10% | 0.26 | 0.55 | 0.18 | $3.0\times$ / $2.7\times$ |
| $\text{TRADES}_{\lambda=\frac{1}{6}}$ + $\text{BT}_{N_R=2}$ | **85.89%** | **56.35%** | 0.28 | 0.54 | 0.18 | $2.3\times$ / $2.2\times$ |

Table 3: Robust accuracy and speedup of applying BulletTrain (BT) to MART adversarial training against the white-box $\text{PGD}^{20}$ attack on CIFAR-10 — $\epsilon = 0.8$ is used for MART with BT.

| Defenses | Clean Acc. | Robust Acc. | $\bar{F}_B$ | $\bar{F}_R$ | $\bar{F}_O$ | Theor./Wall-clock Speedup |
|---|---|---|---|---|---|---|
| MART | 84.17% | 57.39% | 1 | 0 | 0 | $1\times$ / $1\times$ |
| MART + FAST | 93.95% | 0.20% | 1 | 0 | 0 | $5.5\times$ / $4.0\times$ |
| MART + $\text{BT}_{N_R=1}$ | 87.12% | 58.11% | 0.29 | 0.44 | 0.27 | $2.5\times$ / $2.1\times$ |
| MART + $\text{BT}_{N_R=2}$ | **86.60%** | **58.74%** | 0.30 | 0.43 | 0.26 | $2.2\times$ / $1.9\times$ |

of BulletTrain remains comparable to TRADES against stronger attacks. Specifically, TRADES with and without BulletTrain $_{N_R=2}$ achieve 54.32% and 54.33% robust accuracy against PGD-100 with 10 restarts, and 54.41% and 54.49% robust accuracy against PGD-1000 with 5 restarts.

TRADES can make trade-offs between the clean and robust accuracy by varying the value of hyperparameter $\lambda$. When aiming for a lower robust accuracy, BulletTrain can further reduce the $N_R$ from two to zero instead of using a smaller $\lambda$ and still obtain similar clean and robust accuracy as $\text{TRADES}_{\lambda=1}$. As a result, BulletTrain achieves $2.3\times$ and $3.0\times$ computation reduction over the TRADES baseline targeting at 56.3% and 49.3% robust accuracy, respectively. In addition to TRADES, we also demonstrate the effectiveness of BulletTrain on MART as listed in Table 3. BulletTrain with $N_R = 2$ is able to improve the clean and robust accuracy by 2.4% and 1.3% with $2.2\times$ less computation compared to the original MART scheme. Similar to TRADES, we can trade off robust accuracy for clean accuracy and speedup by using a smaller $N_R$. When $N_R = 1$, BulletTrain still achieves both higher clean and robust accuracy than the MART baseline and reduces the computational cost by $2.5\times$.

To compare with other efficient methods on adversarial training, we directly apply YOPO (Zhang et al., 2019a) and FAST (Wong et al., 2020) on TRADES and MART. In Table 2, despite the YOPO-2-5 can improve the runtime by $3.1\times$, the robustness accuracy of YOPO-2-5 is 10% lower than the TRADES baseline. In addition, the single-step FAST training algorithm also fails to obtain a robust DNN model against PGD attack. GradAlign (Andriushchenko and Flammarion, 2020) proposes to address the catastrophic overfitting problem of FAST by introducing an additional gradient alignment regularization. While GradAlign can significantly improve the robustness of the model under stronger attacks, we find that combining GradAlign and FAST still does not improve the accuracy of the robustness of FAST. We further compare the effectiveness of BulletTrain and a multi-step variant of FAST in Table 4. For BulletTrain, we change the value of $N_B \in [3, 6]$ while fixing $N_R = 2, N_O = 0$,

Table 4: Comparison of accuracy and theoretical speedup of a multi-step FAST and BulletTrain (BT) on CIFAR-10 — The numbers in parentheses show accuracy differences compared to $\text{TRADES}_{\lambda=1/6}$.

| | TRADES + Multi-step FAST | | | | TRADES + BT | | |
|---|---|---|---|---|---|---|---|
| $N$ | Clean Acc. (%) | Robust Acc. (%) | Theor./Wall-clock Speedup | $N_B$ | Clean Acc. (%) | Robust Acc. (%) | Theor./Wall-clock Speedup |
| 2 | 85.43 $_{(+0.51)}$ | 50.56 $_{(-6.05)}$ | $3.7\times$ / $3.0\times$ | 3 | **86.66** $_{(+1.74)}$ | **55.93** $_{(-0.68)}$ | $3.7\times$ / $3.0\times$ |
| 3 | 84.47 $_{(-0.45)}$ | 53.52 $_{(-3.09)}$ | $2.8\times$ / $2.5\times$ | 4 | 86.17 $_{(+1.25)}$ | 55.77 $_{(-0.84)}$ | $3.3\times$ / $2.8\times$ |
| 4 | 84.23 $_{(-0.69)}$ | 53.95 $_{(-2.66)}$ | $2.2\times$ / $2.2\times$ | 5 | **85.89** $_{(+0.97)}$ | **56.17** $_{(-0.44)}$ | $3.0\times$ / $2.7\times$ |
| 5 | 84.12 $_{(-0.82)}$ | 54.45 $_{(-2.16)}$ | $1.8\times$ / $1.8\times$ | 6 | 86.28 $_{(+1.36)}$ | 56.0 $_{(-0.61)}$ | $2.8\times$ / $2.6\times$ |

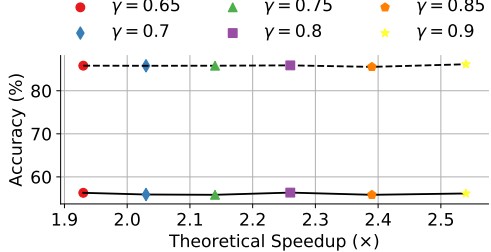
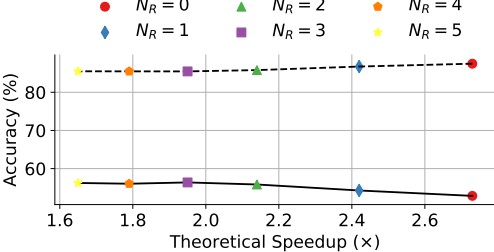

Figure 6: The theoretical speedup vs. accuracy of TRADES+BT under different $\gamma$ values — The dash line shows the clean accuracy and the solid line represents the robust accuracy.

Figure 7: The theoretical speedup vs. accuracy of TRADES+BT under different $N_R$ values — The dash line shows the clean accuracy and the solid line represents the robust accuracy.

and $\gamma = 0.75$ to obtain different computation costs and accuracy. Compared to the multi-step variant of FAST, BulletTrain dominates in clean accuracy, robust accuracy, and theoretical speedup. When the goal is to reduce the computational effort by $3\times$, BulletTrain achieves 1.4% and 2.2% improvement in clean and robust accuracy, respectively. We believe that BulletTrain outperforms existing efficient robust training approaches because BulletTrain can allocate computation for each sample dynamically based on the sample's importance instead of reducing computation for all samples equally,

**Wall-clock speedup.** To measure the wall-clock speedup of BulletTrain, we benchmark the robust DNN training algorithms with and without BulletTrain using a single NVIDIA GPU. We ensure that no other intensive processes are running in parallel with the robust training job. BulletTrain reduces the runtime of AugMix from 1.82 hours to 1.08 hours and from 1.84 hours to 1.10 hours for CIFAR-10 and CIFAR-100, respectively. The runtime is reduced by **1.7×** given the $1.8\times$ theoretical speedup. TRADES baseline is trained with 76 epochs using 44.78 hours on CIFAR-10. BulletTrain is able to reduce the training time of TRADES by **2.2×** compared to the theoretical speedup of $2.3\times$. Moreover, as shown in Table 4, BulletTrain achieves a wall-clock speedup similar to that of FAST when their theoretical speedups are close, suggesting that BulletTrain can be effective in achieving real-world performance gains on the GPU.

**Sensitivity of hyperparameters.** We further investigate the impact of hyperparameters $N_R$ and $\gamma$ on the accuracy and theoretical speedup. Figure 6 shows that the accuracy and theoretical speedup of BulletTrain under different $\gamma \in [0.65, 0.9]$. The clean and robust accuracy are not sensitive to the choice of $\gamma$, showing the stability of BulletTrain. On the other hand, $N_R$ controls the trade-off between the robust accuracy and clean accuracy/theoretical speedup. A smaller $N_R$ improves the clean accuracy and the speedup while reducing the robust accuracy. Unlike FAST, choosing different $N_R \in [0, 5]$ for BulletTrain does not lead to a significant decrease in robust accuracy.

## 5 Related Work

**Robustness against common corruptions.** Data augmentation has been a common technique to improve the generalization of DNNs (Devries and Taylor, 2017; Zhang et al., 2018). Hendrycks and Dieterich (2019) provide benchmarks to evaluate the robustness against common corruptions. AutoAgument (Cubuk et al., 2019) proposes to enhance the robustness by training the model with learned augmentation, which still leads to more than 10% accuracy degradation on CIFAR-10-C. Hendrycks et al. (2020) try further improve the robust accuracy using a mix of augmentations to create different versions of corrupted examples and then leverage the JSD loss between clean and corrupted examples to optimize the model. BulletTrain can reduce the overhead of robust training by only augmenting a subset of examples and computing the JSD loss for those augmented examples.

**Adversarial robustness.** To train a robust network against powerful adversarial attacks, Madry et al. (2018) propose to introduce adversarial examples obtained by using multi-step FGSM during training, namely PGD adversarial training. Recently, PGD adversarial training has been augmented with better surrogate loss functions (Ding et al., 2020; Wang et al., 2020; Zhang et al., 2019b). In particular, MMA training (Ding et al., 2020) attempts to improve the robustness by generating adversarial examples with the optimal perturbation length, where the optimal length $\epsilon^*$ is defined as the shortest perturbation that can cause a misclassification. BulletTrain bears some similarities with the MMA training in the sense that BulletTrain also decides the number of steps based on the margin

between the input samples and the current decision boundary. However, BulletTrain aims to reduce the computation cost of adversarial training by setting an appropriate number of steps for each sample dynamically while MMA requires an expensive bisection search to find the approximate value of $\epsilon^*$.

**Importance sampling.** Importance sampling has been used to speed up DNN training. Schaul et al. (2015), Loshchilov and Hutter (2015), and Chang et al. (2017) sample the data based on the importance metric such as loss. Calculating the loss requires forward propagation, which makes it infeasible to obtain the up-to-date value for each sample. Instead, previous works keep track of the losses of previously trained samples and select important samples based on historical data. However, since the model is updated after each training iteration, the loss of the samples which have not been used recently may be stale. In contrast, BulletTrain still goes through every sample and obtains the up-to-date scores for all samples before allocating the computational cost for each sample.

## 6  Conclusion and Future Work

We introduce a dynamic boundary example mining technique for accelerating robust neural network training, named BulletTrain. Experimental results show that BulletTrain provides better trade-offs between accuracy and compute compared to existing efficient training methods. Potential future work includes determining the number of steps necessary for each sample.

## 7  Acknowledgements

We would like to thank the anonymous reviewers for their constructive feedback on earlier versions of this manuscript. Weizhe Hua and Yichi Zhang are supported in part by the National Science Foundation under Grant CCF-2007832. Weizhe Hua is also supported in part by the Facebook fellowship.

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
