# OpenReview forum: "BulletTrain: Accelerating Robust Neural Network Training via Boundary Example Mining"
_NeurIPS.cc/2021/Conference — NeurIPS 2021 Poster_

### Official Review · Reviewer_d8Mu · 2021-07-02

**Rating:** 6
**Confidence:** 3

**Summary:**

This paper proposes to accelerate robust training by identifying three categories of examples in each mini-batch: outlier examples (the clean example is misclassified), boundary examples (the clean example is correctly classified but can be perturbed to an adversarial example), and robust examples (can be robustly classified). The proposed method sets different numbers of attack steps for the three categories respectively, and in particular, the step number is set to 0 for outlier examples and 2 for robust examples, which is smaller than boundary examples. Thereby, the cost on outlier examples and robust examples is reduced. Experiments show that the proposed method has an around 1.7X~2.1X speed-up with similar robust accuracy.

**Limitations And Societal Impact:**

There seems to be no significant negative social impact.

**Main Review:**

This paper is mostly clearly written, and the proposed method is mostly well motivated. However, I am not certain on some technical details. For line 172, why should outliers be determined based on SVar? If outliers are just misclassified examples, why can’t they be determined just by the classification on the clean examples? Besides, Eq. (6) needs more explanations. Why is this transformation good for separating $X_B$ and $X_R$? Does it really perform well in experiments? In the experiments, there is a speed-up compared to the baseline, but ~2X speed-up is probably not very significant.

I think the comparison with prior works is probably not sufficient, especially on the comparison with prior works for accelerating adversarial training. This paper has mentioned Andriushchenko and Flammarion, 2020; Shafahi et al., 2019; Wong et al., 2020. But the experiments seem to have only involved Wong et al., 2020. I think it is important to also include empirical comparison with other prior works, considering ~2X speed-up is not very large. And a comparison with Zhang el. al., 2019 (cited below) is probably also needed.

Zhang, D., Zhang, T., Lu, Y., Zhu, Z., & Dong, B. (2019). You only propagate once: Accelerating adversarial training via maximal principle. arXiv preprint arXiv:1905.00877.


====

Update after rebuttal

Thanks to the authors for the rebuttal! I think most of my concerns have been well addressed. With the new discussion and results from the authors, I'm increasing my recommendation to 6.

**Time Spent Reviewing:**

3

---

> ### Author Response · Authors · 2021-08-10
> **Response to Reviewer d8Mu**
>
> Thank you for your insightful comments and suggestions. We will include more results of comparison with previous art in the final version. Below are detailed responses to questions.
>
> 1. **Determine outliers based on SVar.**
>
> 	Sorry for the confusion. The outliers are determined by the classification of clean examples, which is the sign of the SVar. We combine the sign of the prediction with the predicted variance because we want to write a unified classification rule for all examples, as in Equation (7).
>
> 2.  **The rationale and effectiveness of the proposed method for separating examples.**
>
> 	$F_R$ represents the estimated fraction of robust examples (i.e., robust accuracy). Since the robustness of the model does not improve significantly over several iterations, the robust accuracy of adjacent mini-batches should be reasonably consistent. Therefore, we can obtain less noisy estimates of $F_R$ by using the moving average of $F_R$ measured in previous iterations. In the literature, batch normalization also uses moving averages and variances as a proxy for actual batch averages and variances in the inference process.
> 	\
> 	\
> 	We validated the effectiveness of the proposed scheme through two studies. We first tested TRADES with BulletTrain using an “oracle” separation (i.e., 10-step PGD) between examples. The robust accuracy with “oracle” separation is 56.73% on CIFAR-10, which is only 0.4% higher than using the signed prediction variance. In addition, we also measured the classification accuracy of separating $X_O$, $X_R$, and $X_B$ over the entire 76 epochs of TRADES adversarial training, where ground truth labels were generated using a 10-step PGD. Using the proposed moving average of robust accuracy (i.e., $F_R$) achieves 83.6% classification accuracy, which is 1.5% higher than using kernel density estimation.
>
> 3. **2X speed-up is not very significant.**
>
> 	It is important to note that the 2x wall-clock speedup is obtained on the state-of-the-art robust training methods such as AugMix, TRADES, and MART, while most of the previous work only improved the runtime of the original PGD training [C1] that achieves a subpar robust accuracy. Since TRADES and MART require computing the gradient with respect to the weights using both clean and adversarial examples, the computational cost of the model update step is doubled. Thus, even if we reduce the number of steps to generate adversarial examples to two (the 1-step version of TRADES and MART are not robust), the wall-clock speedup is limited to 3x. It is also worth noting that TRADES+BT$_{N_B=3}$ in Table 4 can also achieve 3x actual speedup with only 0.7% degradation in robust accuracy.
>
> 4. **Comparison with prior works is not sufficient.**
>
> 	To make a fair comparison between BulletTrain and previous work, we combine GradAlign (Flammarion et al., 2020), Free (Shafahi et al., 2019), and YOPO (Zhang et al., 2019) with TRADES. The network architecture is WideResNet-34-10, and all hyperparameters are set to the same as the TRADES baseline. The experimental results show that all previous works lead to a significant degradation in robust accuracy, while BulletTrain can achieve a 2-3 times speedup with a negligible decrease in robust accuracy. The details of the experiments are as follows:
> 	\
> 	\
> 	We ran YOPO using the released code and the hyperparameter settings from the original paper (m=3/n=4 and m=2/n=5). Although YOPO-2-5 is able to improve the runtime by 3.1 times, the robust accuracy of YOPO-2-5 is 10% lower than the TRADES baseline.
> 	\
> 	\
> 	Free can obtain the gradient with respect to the adversarial examples and weights using only one forward and backward propagation, as it uses cross-entropy loss for both purposes. However, TRADES achieves high robust accuracy by using different loss functions for adversarial example generation and model updating. Therefore, applying Free on TRADES is 2 times more expensive than applying it on PGD. Due to the inefficiency of applying Free on TRADES, both Free-2 and Free-3 achieve less than 2x speedup. Specifically, TRADES$_{\lambda=1/12}$+Free-3 archives 5% lower robust accuracy with only 1.3x speedup compared to the TRADES baseline.
> 	\
> 	\
> 	GradAlign was proposed to address the catastrophic overfitting problem of FAST under stronger attacks (i.e., large $\epsilon$) by introducing extra gradient alignment regularization. While GradAlign can significantly improve the robustness of the model under stronger attacks, the robust accuracy remains roughly the same under a common L infinity attack with $\epsilon$ = 8/255. Similarly, we find that combining GradAlign and FAST still does not improve the robust accuracy of FAST.
>
> | Defenses | Clean Acc. (%) | Robust Acc. (%) | Wall-clock Speedup ($\times$) |
> |:---:|:---:|:---:|:---:|
> |  TRADES$\_{\lambda=1/6}$+BT$_{N_R=2,N_B=10}$ | 85.89| 56.35 | 2.2
> | TRADES$\_{\lambda=1/6}$+BT$_{N_R=2,N_B=3}$ | 86.66 | 55.93 | 3.0
> | TRADES$_{\lambda=1}$+YOPO-2-5 | 88.47 | 45.28 | 3.1
> | TRADES$_{\lambda=1/6}$+YOPO-2-5 | 89.95 | 46.60 | 3.1
> | TRADES$_{\lambda=1}$+YOPO-3-4 | 93.64 | 1.23 | 2.7
> | TRADES$_{\lambda=1/6}$+YOPO-3-4 | 93.62 | 2.16 | 2.7
> | TRADES$_{\lambda=1/6}$+Free-2 | 91.53 | 45.34 | 1.9
> | TRADES$_{\lambda=1/12}$+Free-2 | 90.00 | 49.75 | 1.9
> | TRADES$_{\lambda=1/6}$+Free-3 | 89.44 | 48.36 | 1.3
> | TRADES$_{\lambda=1/12}$+Free-3 | 87.26 | 51.55 | 1.3
> | TRADES$_{\lambda=1/6}$+FAST+GradAlign | 94.10 | 5.30 | -
>
> [C1] Madry et al., “Towards Deep Learning Models Resistant to Adversarial Attacks ”,  ICLR’18.

---

### Official Review · Reviewer_5TSA · 2021-07-15

**Rating:** 6
**Confidence:** 4

**Summary:**

The paper proposes a technique for reducing the computational cost of robust training by dynamically selecting which samples to assign the most compute power to during either adversary generation (for robustness against adversaries) or the forward pass of alternative samples (for robustness against common corruptions). The approach assigns compute power according to proximity to decision boundary; such quantity is estimated, in turn, by a (simple yet effective) proxy that is defined in the paper. The method is sensible, intuitive and well motivated. Experimental results show sizable gains in efficiency with little or no losses in robustness.

**Limitations And Societal Impact:**

Yes, they did.

**Main Review:**

Strengths:
+ The paper's object of study is interesting for the robustness community (in terms of providing further understanding as to what are the particular factors that affect robustness) and also for the deep learning community (in terms of how such an approach can make robust training more compute friendly for other applications).
+ The approach presented in the paper is sensible, well motivated and intuitive.
+ The design decisions are also sensible and are reasonably validated experimentally.
+ The experimental results show consistent gains in efficiency with little or no losses in performance.
+ The paper's writing is adequate, and the narrative helps the reader understand the thought process behind the approach.
+ While sharing motivation/intuitions with previous works, the paper's proposal in itself is original.


Weaknesses that affect my rating:
Comment: overall I like the paper's idea, but the main drawback from my perspective is that there are a couple unclear items in the experiments. In particular, the largest one being the disproportionate importance that *theoretical* speedups are given, instead of actual empirical speedups. In any case, I think the empirical results would still yield the same conclusions as those presented in the paper, but these results should be reported. If these results are reported and the conclusions are still the same, as I suspect (and provided other reviewers do not raise concerns I may have overlooked), I am willing to suggest the acceptance of this paper. Next, I outline my largest concerns (in order of importance):
+ In Tables 1-4, what does "Theor. Speedup" mean? Are these *theoretical* gains, or actual gains as measured empirically? If they are the former (as I assume, given the paragraph named "Wall-clock speedup" on L258), then it must also be explained how the fractions $\bar{F}_x$ were estimated across training iterations, as they are fundamental for the overall compute time, right? Also importantly, if all the clean/robust accuracies were computed such that these tables could be filled, it also means that the actual speedups must have been measured too, so why not report them in these tables along with the theoretical speedups? (and the same comment can be made for Figs. 6 and 7)
+ Why are not results for MART reported in the "Wall-clock speedup" section? These results must have been computed for filling Table 3, right?
+ I think the experiment reported in the Section 3.1 (more specifically L132) is interesting, and I agree with the results and the conclusion drawn from them. However, I would inquire about an implementation concern: Figure 2, in particular subfigures (b) and (c), shows that there is a *dramatic* change in robust accuracy when varying steps in for $N_B$ from $1\to2\to3$: robust accuracy can suddenly jump from $\sim0$ to $>0.8$. I find this result intriguing, and even a bit suspicious. Is it possible that this is because the size of steps taken by PGD are not dependent on the amount of steps that are taken (as this is nowhere reported for this experiment, but apparently was indeed used for the main experiments, as reported in L228)? That is, can it be the case that we are observing this phenomenon mainly because training with such few steps does not allow optimization to reach the boundary of the $\epsilon$-ball constraints? In principle this would mean that the model is being trained against weaker adversaries, while always being evaluated against stronger ones. If this is the case, I think this invalidates the results presented here (even though I still think that fixing for this issue would still yield the same overall conclusions presented in the paper).
+ The distribution reported in Figure 5(a) seems a bit weird: (i) why are these results for a single mini-batch? Why not show this for the entire dataset? (ii) at what stage of training is this being computed? Please specificy, and (iii) a large portion of the examples appear to be incorrectly classified ("outliers"), but this contradicts what was shown in Figure 3, no?
+ Did the authors consider assessing how good of an estimate is the usage of Signed-Prediction Variance as a proxy for knowing which samples are $\mathbf{x}_R$ or $\mathbf{x}_B$? For instance, computing this quantity for all samples and also computing the adversaries of these samples with a PGD attack with lots of iterations, or running DeepFool/FAB to estimate the distance to the decision boundary? Such validation would have been interesting.
+ The intuition behind using the consistency of robust accuracy across adjacent mini-batches is sensible, but would not this assumption (namely Eqn. (6)) induce a bias for $F_R$? As the algorithm will always be estimating the new $F_R$ by using the PGD steps budget that was itself defined by the previous $F_R$? Could the authors comment on this phenomenon?
+ I think the paper could benefit from commenting about its relation with the following:
- Attacks Which Do Not Kill Training Make Adversarial Learning Stronger, Zhang et al., ICML2020
- Hold me tight! Influence of discriminative features on deep network boundaries, Ortiz-Jimenez et al., NeurIPS2020
- Decision Boundary Analysis of Adversarial Examples, He et al., ICLR2018
- Improving Adversarial Robustness Through Progressive Hardening, Sitawarin et al., 2020.



Weaknesses that do not affect my rating but should be addressed if possible:
+ L58: the second $x_i$ in the line is not boldface. Is this on purpose?
+ Algorithm 1 and 2: nit-picky comment: use the labels $y_i$, probably in the $G_N$ and $\mathcal{L}$ functions.
+ L133: unsure about the reference for MNIST.
+ Figure 4, plot x-label legend: iteraions
+ Figure 5(a). x-axis legend: predidction
+ Algorithm 2, L8: I think $F_O$ is never used.
+ Table 1 and 2 should report in the caption what $\bar{F}_x$ stands for. Further, how exactly are these quantities being computed? By the end of training? Or when?
+ In Tables 1-4, what convention do bold-faces and underlines follow? I think their current usage is misleading, as the reader may think the best results are obtained with the proposed approach, when they are not.
+ The caption of Table 2 states that $\epsilon = 0.8$ is used for TRADES with BT (same thing for Table 3 and MART). What does this experimental setup mean? That the TRADES adversaries are being computed within a ball of this radius? Or that the robust accuracy reported here is being computed within such ball? The answer to this question could very much change how the reported results are understood.
+ L230: with no and
+ L263: baseline are
+ Figure 6 caption: dash line



**Time Spent Reviewing:**

6

---

> ### Author Response · Authors · 2021-08-10
> **Response to Reviewer 5TSA**
>
> Thank you for your insightful and very detailed comments. We will include the wall-clock speedup of all experiments in the final version. In addition, we will discuss related work and improve the writing of the paper for the final version based on the suggestions. Here are the detailed responses to the questions.
>
> 1. **Definition of theoretical speedup and the wall-clock speedup of all experiments.**
>
> 	The theoretical speedup is calculated using the measured average fraction of $X_B$, $X_O$, and $X_R$ (i.e., $F_B$, $F_O$, and $F_R$ ), assuming that all computations without dependencies can be parallelized with unlimited hardware resources. Specifically, for TRADES, the ten iterations (i.e., one forward and backward propagation is considered as one iteration) for generating adversarial examples and one iteration for updating the model must be executed sequentially. Therefore, we can write theoretical speedup as follows:
>
> 	Theoretical speedup = ($F_B \times N_B + F_R \times N_R + F_O \times N_O+1$)/(10 + 1)
> 	\
> 	\
> 	We did not report the wall-clock speedup for all experiments upon submission as many of them are executed on different GPUs, so the absolute run times are not comparable. We now obtain the wall-clock speedups for all experiments in Table 1-4 using a single Nvidia 2080 Ti GPU and list them along with the theoretical speedup. As listed in the table below, BulletTrain ($N_B$=[3...6]) is able to obtain a real speedup similar to FAST ($N$=[2...5]) under the same theoretical savings, which proves the effectiveness of BulletTrain.
>
> 	| Defenses | Dataset | Theor. Speedup ($\times$) | Wall-clock Speedup ($\times$)|
> 	|:---:|:---:|:---:|:---:|
> 	| AugMix+BT | CIFAR-10 | 1.8 | 1.7 |
> 	| AugMix+BT | CIFAR-100 | 1.8 | 1.7 |
>
> 	| Defenses | Robust Acc. (%) | Theor. Speedup ($\times$) | Wall-clock Speedup ($\times$) |
> 	|:---:|:---:|:---:|:---:|
> 	|  TRADES$\_{\lambda=1/6}$+FAST | 4.48|5.5 | 3.7 |
> 	| TRADES$\_{\lambda=1/6}$+BT$_{N_R=0}$ | 52.10 | 3.0 | 2.7 |
> 	| TRADES$\_{\lambda=1/6}$+BT$_{N_R=2}$ | 56.35 | 2.3 | 2.2 |
>
> 	| Defenses | Robust Acc. (%) | Theor. Speedup ($\times$) | Wall-clock Speedup ($\times$) |
> 	|:---:|:---:|:---:|:---:|
> 	|  MART$\_{\lambda=1/6}$+FAST | 0.20 | 5.5 | 4.0 |
> 	| MART$\_{\lambda=1/6}$+BT$_{N_R=1}$ | 58.11 | 2.5 | 2.1 |
> 	| MART$\_{\lambda=1/6}$+BT$_{N_R=2}$ | 58.74 | 2.2 | 1.9 |
>
>
> 	| $N$ (FAST)| Robust Acc. (%) | Theor. Speedup | Wall-clock Speedup | $N_B$ (BT)|  Robust Acc. (%) | Theor. Speedup | Wall-clock Speedup |
> 	|:---:|:---:|:---:|:---:|:---:|:---:|:---:|:---:|
> 	| 2 | 50.56 | 3.7$\times$ | 3.0$\times$ | 3 | 55.93 | 3.7$\times$ | 3.0$\times$ |
> 	| 3 | 53.52 | 2.8$\times$ | 2.5$\times$ | 4 | 55.77 | 3.3$\times$ | 2.8$\times$ |
> 	| 4 | 53.95 | 2.2$\times$ | 2.2$\times$ | 5 | 56.17 | 3.0$\times$ | 2.7$\times$ |
> 	| 5 | 54.45 | 1.8$\times$ | 1.8$\times$ | 6 | 56.00 | 2.8$\times$ | 2.6$\times$ |
>
> 	It is worth noting that when $N_B$ or $N$ is small, the difference between the theoretical speedup and the wall-clock speedup is relatively large. For example, TRADES+FAST$\_{N=2}$ and TRADES+BT$\_{N_B=3}$ only achieve 3x actual speedup given a theoretical speed of 3.7x. The reason is that the update step of TRADES requires using clean and corrupted examples to compute the gradients with respect to the weights, which cannot be fully parallelized on a GPU. When the number of iterations to generate adversarial examples becomes small, the update step will dominate the runtime, thus limiting the actual speedup.
>
> 2. **Wall-clock speedup for MART+BulletTrain.**
>
> 	Listed in the table above.
>
> 3. **Concerns about the implementation of the experiment in Section 3.1.**
>
> 	Thanks for pointing out the caveat in the implementation of the experiment. We set the step size for robust, boundary, and outlier examples separately in the same way as the experiments in Table 4. The step size was set to ($1.7 * \epsilon$/number of steps), which should be large enough to reach the boundary of the $\epsilon$-ball. We further increased the step size to ($2.5 * \epsilon$/number of steps) and still observed a similarly dramatic change in robustness accuracy. We hypothesize that setting $N_B$ to 0 may be the cause of this phenomenon.
>
> 4. **Clarification of Figure 5(a).**
>
> 	Since the signed prediction variance of $X_B$ and $X_R$ increases during training (as shown in Figure 4), it is only meaningful to show the prediction variance of the entire dataset when the variance is calculated in the same iteration. However, the main purpose of Figure 5 is to illustrate how the examples in a mini-batch are divided into three classes (i.e., the core part of BulletTrain) and to show that $X_B$ and $X_R$ are not completely separable using the signed prediction variance. This batch of data was obtained from the $800^{th}$ iteration of CIFAR-10 adversarial training (each epoch contains 500 iterations). We chose to visualize the early phase of training intentionally because it contains a sufficient number of $X_B$, $X_R$, and $X_O$. As the adversarial training on CIFAR-10 converges slower than the training on MNIST (Figure 3), there are still a large number of misclassified examples.
>
> 5. **Evaluate the effectiveness of using the signed prediction variance.**
>
> 	We conducted the proposed study for selecting the metrics and prediction methods. In addition to the proposed method, we also tried to use gradient norm for estimating the uncertainty and kernel density estimation (KDE) for separating $X_R$ and $X_B$.
> \
> \
> 	We measured the classification accuracy of separating $X_O$, $X_R$, and $X_B$ over the entire 76 epochs of TRADES adversarial training, where ground truth labels were generated using a 10-step PGD. Using the proposed moving average of robust accuracy (i.e., $F_R$) achieves 83.6% classification accuracy, which is 1.5% higher than using KDE. The difference in accuracy between using the signed prediction variance and gradient norm is negligible. In addition, we also tested TRADES with BulletTrain using an “oracle” separation (i.e., 10-step PGD) between examples. The robust accuracy with “oracle” separation is 56.73% on CIFAR-10, which is only 0.4% higher than using the signed prediction variance.
>
> 6. **Bias in estimating $F_R$.**
>
> 	Theoretically, the proposed scheme is likely to overestimate $F_R$ since the $F_R$ of previous iterations is measured using the predicted number of PGD steps and $F_R$ generally increases during the training process. To compensate for the overestimation, we introduce a hyperparameter gamma in Equation (6) to estimate $F_R$ more conservatively. We found that setting gamma to be less than 1 (e.g., 0.8) helps to restore the robust accuracy of the baseline in our empirical evaluations.
>
> 7. **Comment on the related works.**
>
> 	Thanks for pointing out these closely related works. First, we believe both BulletTrain, friendly adversarial training [C1], and progressive hardening [C4] made a similar observation that adversarial training only requires corrupted examples with appropriate difficulty. Friendly adversarial training and progressive hardening leverage the intuition to improve the robust and clean accuracy tradeoff whereas BulletTrain aims at speedup the adversarial training. The related works in [C2, C3] provide some intriguing insights into the relationship between examples and decision boundaries. Especially, Hold me tight [C2] pointed out that the success of adversarial training arises from hiding some discriminative features from the model. BulletTrain is related to [C2] as it focuses on hiding the discriminative features in boundary examples, which are key examples containing rich features from which the model is learning.
>
> [C1] Attacks Which Do Not Kill Training Make Adversarial Learning Stronger, Zhang et al., ICML’20.
>
> [C2] Hold me tight! Influence of discriminative features on deep network boundaries, Ortiz-Jimenez et al., NeurIPS’20.
>
> [C3] Decision Boundary Analysis of Adversarial Examples, He et al., ICLR’18.
>
> [C4] Improving Adversarial Robustness Through Progressive Hardening, Sitawarin et al., ArXiv’20.

---

> > ### Comment · Reviewer_5TSA · 2021-08-29
> > **Follow-up on authors' response**
> >
> > Thank you for your responses.
> >
> > I think most of the individual reviewers' concerns have been addressed. I see you provided further experiments that make the paper's case stronger. For instance, I see more detailed comparisons with other methods both in terms of robustness performance and speedups (e.g. comparison with YOPO and Free), which was one of the main issues raised by the reviewers. Regarding my concerns in particular, I think they have been mostly addressed.
> >
> > Thus, I will upgrade my rating to 6.

---

### Official Review · Reviewer_jFeM · 2021-07-16

**Rating:** 6
**Confidence:** 4

**Summary:**

This paper introduces BulletTrain - an algorithm for speeding up robust training algorithms for both adversarial and corruption based robustness. This is an important problem to study because robust training (adversarial training in particular) is generally expensive which makes it hard to scale these algorithms on large datasets. The algorithm proposed in this paper, while being lightweight and easy to implement, shows strong empirical gains. Overall, I like this paper.

**Limitations And Societal Impact:**

As pointed out in the previous section, I see the following limitations:

- Deciding (N_R, N_o, N_b) might be a laborious task
- Experimental results on stronger attacks like PGD-100/1000 are missing
- Adversarial training on large scale datasets Imagenet would have really helped showcase the efficacy of this method

**Main Review:**

The main contribution of this paper is to divide the training examples into three sets: outliers, boundary and robust examples. After dividing these samples, each set of samples is allotted a different perturbation generation budget. While dividing samples into different sets have been explored already in the context of adversarial training (for example, [1]), I believe this paper does a good job of neatly packaging this idea into a simple framework. The authors argue that boundary samples are the most important ones to train on. This has been neatly illustrated in Figure 2. Use of lower budgets for robust and outlier samples provides good gains in computation. To separate the three sets of samples, the authors use a metric called signed prediction variance which is again very easy to compute. Good empirical results on different benchmarks highlight the significance of the proposed approach.

Some comments:

I’m not sure if outliers is a good term to be used for misclassified samples.

Can you comment on the failure modes of signed prediction variance. I agree that prediction variance is a decent measure of uncertainty, but it has some drawbacks as well. Can you use other measures used in uncertainty estimation to improve the performance?

How to decide N_R, N_o and N_b? Is there a principled way to decide this? For example if I give you a computational budget, how do you choose  (N_R, N_o, N_b) triplet for this budget?

What happens if you perform adversarial evaluation on stronger attacks, say PGD-100 or PGD-1000 with multiple restarts? Its important to evaluate stronger attacks to check if things like gradient masking happens. I would recommend authors to have these numbers to get a complete picture.

It would be really nice if the authors perform adversarial training on large scale datasets like Imagenet where adversarial training is super expensive. If BT is able to train adversarial training on Imagenet faster, it would be an amazing result.

I appreciate the authors for comparing wall clock speedup with theoretical speedup and showing that they both compare.

[1] Balaji et al., “Instance adaptive adversarial training: Improved accuracy tradeoffs in neural nets.”


**Time Spent Reviewing:**

3 hours

---

> ### Author Response · Authors · 2021-08-10
> **Response to Reviewer jFeM**
>
> Thank you for your insightful comments and suggestions. We will include the results of the stronger attacks on BulletTrain in the final version. Here are the detailed responses to the questions.
> 1. **Failure modes and effectiveness of using signed prediction variance.**
>
> 	The main failure mode of using the signed prediction variance is that the robust and boundary samples are not fully separable as shown in Figure 5(a). Even though the examples cannot be perfectly classified using the prediction variance, BulletTrain achieves robust accuracy that is very close to the "oracle" method. On CIFAR-10, the robust accuracy obtained using TRADES with an "oracle" separation between samples (i.e., 10-step PGD) was 56.73%, only 0.4% higher than that using the signed prediction variance. We also measured the classification accuracy of separating $X_B$, $X_O$ and $X_R$ over the entire 76 epochs of TRADES adversarial training on CIFAR-10. Using signed prediction variance can achieve an average accuracy of 83.6%, where ground truth labels are generated using a 10-step PGD.
> 	\
> 	\
> 	In the literature on importance sampling, both prediction variance and gradient norm are top candidates for uncertainty estimation. Since our goal is to speed up robust training, we need a computationally efficient way to estimate uncertainty. We try to separate examples using the prediction variance and the upper bound on the gradient norm derived in [C1], both of which require only one forward propagation. We find that the difference in performance using these two methods is negligible and decide to use the signed prediction variance as it is easier to obtain.
> 2. **Decide $N_R$, $N_O$ and $N_B$ for BulletTrain.**
>
> 	We agree that this is a very interesting research question. Currently, for all adversarial training algorithms, the number of iterations is decided empirically. For example, N is empirically set to be 7 and 10 in PGD and TRADES training, respectively. In this work, we found that 0 and 2 iterations are usually sufficient for misclassified and robust samples on MNIST and CIFAR datasets in our experiments. Given a computational budget, we can first fix $N_O$ and $N_R$ to 0 and 2, and then set $N_B$ accordingly.
> 3. **PGD-100 and PGD-1000 with multiple restarts.**
>
> 	The following table shows the robust accuracy of TRADES with and without BulletTrain using a stronger PGD-100 with 10 restarts and PGD-1000 with 5 restarts. The step size of the PGD-100 and PGD-1000 attack is set to be ($2.5*\epsilon$/number of steps) as suggested in [C2].
>
> 	| Defenses | PGD-20 (no restarts) | PGD-100 (10 restarts) | PGD-1000 (5 restarts) |
> 	|:---:|:---:|:---:|:---:|
> 	| TRADES$\_{\lambda=1/6}$ | 56.61 | 54.32 | 54.41 |
> 	| TRADES$\_{\lambda=1/6}$+BT$_{N_R=0}$ | 52.10| 49.04 | 49.24 |
> 	| TRADES$\_{\lambda=1/6}$+BT$_{N_R=2}$ | 56.35 | 54.33 | 54.49 |
>
>
> 4. **Adversarial training on large-scale datasets like ImageNet.**
>
> 	We agree that it is important to accelerate adversarial learning on large datasets like ImageNet, but this often takes several weeks to complete. We believe that BulletTrain may achieve higher computational savings on ImageNet, as the percentage of misclassified examples on ImageNet will be much higher than on CIFAR-10. However, due to the lack of a baseline (i.e., neither TRADES nor MART were evaluated on ImageNet) and the limited computational resources we had, we were unable to report the ImageNet results by the rebuttal deadline.
>
> [C1] Katharopoulos et al., “Not all samples are created equal: Deep learning with importance sampling”, ICML’18.
>
> [C2] Madry et al., “Towards Deep Learning Models Resistant to Adversarial Attacks ”, ICLR’18.

---

### Author Response · Authors · 2021-08-10
**General response**

We thank all reviewers for their constructive feedback about our work!

We would like to emphasize that although we focused more on demonstrating the speed-up on adversarial training due to its popularity, our method is in fact compatible with more general robust training routines as well. We showed this empirically by using BulletTrain to speed up AugMix training by 1.7x without any loss in robustness to common corruptions. To our best knowledge, prior work on speeding up robust training has mostly focused on the fast generation of adversarial examples, which is not applicable to other robust training techniques such as AugMix. It is arguable that common corruptions represent a more realistic and immediate threat to model integrity, hence our approach may have a larger impact than previously proposed efficient adversarial training methods.

---

### Author Response · Authors · 2021-08-23
**Looking forward to hearing feedback on our response**

We again thank all reviewers for their insightful feedback. In the rebuttal, we believe we have addressed most of the major concerns raised by the reviewers. In particular, we provide the following additional experimental results:

-   Reviewer jFeM: robustness against stronger attacks

-   Reviewer 5TSA: wall-clock speedup for all experiments

-   Reviewer d8Mu: comparison with other acceleration methods for adversarial training


As the deadline for rolling discussions approaches, we would like to hear reviewers' thoughts on our responses and are willing to address and clarify any additional concerns.

---

### Decision · Program_Chairs · 2021-09-27

**Decision:**

Accept (Poster)

**Comment:**

This work proposes to a method to accelerate robust training of neural networks by focusing on boundary examples. The idea is interesting and is well received by the reviewers. The paper is well motivated and clearly writing. Authors' rebuttal also did a great work at resolving the concerns from the reviewers.


On the other hand, there are some weaknesses that we hope the authors make efforts to address in the final version. This includes (as summarized by reviewer 5TSA during discussion): (1) experimental validation of the signed-prediction variance proxy used in the paper, and (2) lack of comparison with other defenses/attacks. In particular, Reviewer d8Mu mentioned critical comparisons that the paper should have reported initially: comparisons with YOPO, Free and GradAlign. Reviewer 5TSA  was particularly concerned with the disproportionate importance given to theoretical speedups in contrast to empirical speedups.